# Neural Networks for Modeling Source Code Edits

## Abstract

Programming languages are emerging as a challenging and interesting domain for machine learning. A core task, which has received significant attention in recent years, is building generative models of source code. However, to our knowledge, previous generative models have always been framed in terms of generating static snapshots of code. In this work, we instead treat source code as a dynamic object and tackle the problem of modeling the edits that software developers make to source code files. This requires extracting intent from previous edits and leveraging it to generate subsequent edits. We develop several neural networks and use synthetic data to test their ability to learn challenging edit patterns that require strong generalization. We then collect and train our models on a large-scale dataset consisting of millions of fine-grained edits from thousands of Python developers. From the modeling perspective, our main conclusion is that a new composition of attentional and pointer network components provides the best overall performance and scalability. From the application perspective, our results provide preliminary evidence of the feasibility of developing tools that learn to predict future edits.

## 1 Introduction

Source code repositories are in a state of continuous change, as new features are implemented, bugs are fixed, and refactorings are made. At any given time, a developer will approach a code base and make changes with one or more intents in mind. The main question in this work is how to observe a past sequence of edits and then predict what future edits will be made. This is an important problem because a core challenge in building better developer tools is understanding the intent behind a developer's actions. It is also an interesting research challenge, because edit patterns cannot be understood only in terms of the content of the edits (what was inserted or deleted) or the result of the edit (the state of the code after applying the edit). An edit needs to be understood in terms of the relationship of the change to the state where it was made, and accurately modeling a sequence of edits requires learning a representation of the past edits that allows the model to generalize the pattern and predict future edits.

As an example, consider Figure 1. We show two possible edit sequences, denoted as History A and History B. Both sequences have the same state of code after two edits (State 2), but History A is in the process of adding an extra argument to the `foo` function, and History B is in the process of removing the second argument from the `foo` function. Based on observing the initial state (State 0) and the sequence of edits (Edits 1 & 2), we would like our models to be able to predict Edit 3. In the case of History A, the specific value to insert is ambiguous, but the fact that there is an insertion at that location should come with reasonably high confidence.

The main challenge in modeling sequences of edits is in how to develop good representations that will both capture the required information about intent, as well as scale gracefully with the length of the sequence. We consider two representations of edits that we call *explicit* and *implicit* representations. The explicit representation explicitly instantiates the state resulting from each edit in the sequence, while the implicit representation instantiates a full initial state and then subsequent edits in a more compact diff-like representation. On the explicit representation we consider a hierarchical recurrent pointer network model as a strong but computationally expensive baseline. On the implicit representation, we consider a vanilla sequence-to-sequence model, and a two-headed attention-based model with a pointer network head for producing edit positions and a content head for producing edit

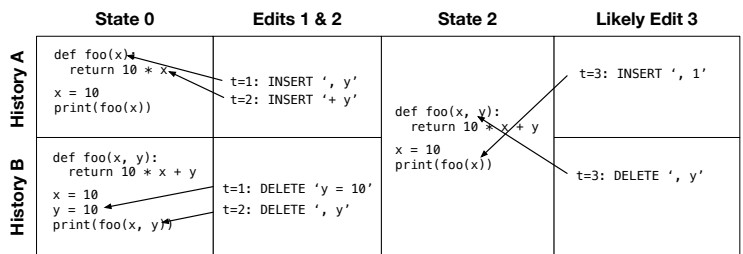

Figure 1: Two illustrative edit sequences. History A and B share the same State 2, but based on the histories, it is more likely that History A will continue by modifying the call to `foo` to take an extra argument and History B will continue by modifying the definition of `foo` to take only one argument.

contents. These models demonstrate tradeoffs that arise from different problem formulations and inform design decisions for future models of edit sequences.

On carefully designed synthetic data and a large-scale dataset of fine-grained edits to Python source code, we evaluate the scalability and accuracy of the models on their ability to observe a sequence of past edits and then predict future edits. We show that the two-headed attention model is particularly well-suited to achieving high accuracy, well-calibrated confidence, and good scalability on the real data, which makes us optimistic about future prospects of developer tools that learn to extract intent from developers as they make edits to large, real code bases. In total, this work formalizes the problem of learning from and predicting edit sequences, provides an initial exploration of model space, and demonstrates applicability to the real-world problem of learning from edits that developers make to source code.

## 2 PROBLEM FORMULATION

**Implicit vs. Explicit Data Representation.** The first question is how to represent edit sequence data. We define two data formats having different tradeoffs. The *explicit* format (Figure 2 (a)) represents an edit sequence as a sequence of sequences of tokens in a 2D grid. The inner sequences index over tokens in a file, and the outer sequence indexes over time. The task is to consume the first $t$ rows and predict the position and content of the edit made at time $t$. The *implicit* format (Figure 2 (b)) represents the initial state as a sequence of tokens and the edits as a sequence of (`position`, `content`) pairs.

The explicit representation is conceptually simple and is relatively easy to build accurate models on top of. The downside is that processing full sequences at each timestep is expensive and leads to poor scaling. Conversely, it is easy to build scalable models on the implicit representation, but it is challenging to recognize more complicated patterns of edits and generalize well. We show experimentally in Section 6 that baseline explicit models do not scale well to large datasets of long sequences and baseline implicit models are not able to generalize well on more challenging edit sequences. Ideally we would like a model that operates on the implicit data format but generalizes well. In Section 4 we develop such a model, and in Section 6 we show that it achieves the best of both worlds in terms of scalability and generalization ability.

**Notation.** A state $s$ is a sequence of discrete tokens $s = (s_0, \ldots, s_M)$ with $s_m \in \mathcal{V}$, and $\mathcal{V}$ is a given vocabulary. An edit $e^{(t)} = (p^{(t)}, c^{(t)})$ is a pair of position $p^{(t)} \in \mathbb{N}$ and content $c^{(t)} \in \{\text{DELETE}\} \cup \mathcal{V}$. An edit sequence (which we also call an *instance*) is an initial state $s^{(0)}$ along with a sequence of edits $e = (e^{(1)}, \ldots, e^{(T)})$. We can also refer to the implied sequence of states $(s^{(0)}, \ldots, s^{(T)})$, where $s^{(t)}$ is the state that results from applying $e^{(t)}$ to $s^{(t-1)}$.

There are two representations for position. In the explicit representation, $p^{(t)} = p_e^{(t)}$ is the index of the token in $s^{(t-1)}$ that $c^{(t)}$ should be inserted after. If $c^{(t)}$ is a DELETE symbol, then $p_e^{(t)}$ is the index of the token in $s^{(t-1)}$ that should be deleted. In the implicit representation, we assign indices $0, \ldots, M$ to tokens in the initial state and indices $M + 1, \ldots, M + T$ to the edits that are made; i.e., if a token is inserted by $e^{(t)}$ then it is given an index of $M + t$. The position $p^{(t)} = p_i^{(t)}$ is the index of the token that the edit should happen after. For example, see Figure 2 (b). The second edit is to

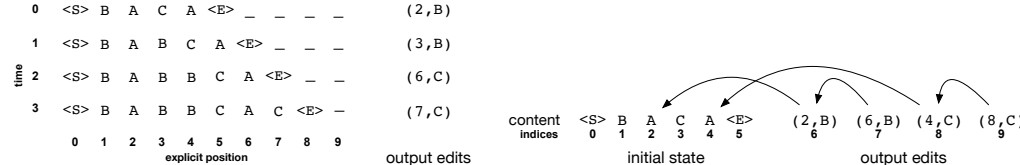

(a) Explicit edit sequence. Left: explicit state at each time. Right: position and content of the edits.

(b) Implicit edit sequence. The position components use different indexing from the explicit representation.

Figure 2: (a) Explicit and (b) implicit representations of a sequence of edits transforming 'BACA' to 'BABBCACC'. $<$S$>$ and $<$E$>$ are special start and end tokens.

insert a B after the B that was inserted in the first edit. The index of the first B is 6, and the position $p_i^{(7)} = 6$, denoting that the second B should be inserted after the first B.

**Problem Statement.** We can now state the learning problems. The goal in the explicit problem is to learn a model that maximizes the likelihood of $e^{(t)}$ given $\boldsymbol{s}^{(0)}, \ldots, \boldsymbol{s}^{(t-1)}$ and the implicit problem is to learn a model that maximizes the likelihood of $e^{(t)}$ given $\boldsymbol{s}^{(0)}, e^{(1)}, \ldots, e^{(t-1)}$ for all $t$.

## 3 BASELINE MODELS

**Baseline Explicit Model.** The baseline explicit model is a two-level Long Short-Term Memory (LSTM) neural network (Hochreiter & Schmidhuber, 1997) similar to hierarchical RNN models like in Serban et al. (2016). We refer to the hidden vector for the $m^{\text{th}}$ token in timestep $t$ as $\boldsymbol{h}^{(t,m)} \in \mathbb{R}^D$. In the simplest version of the model, the first level LSTM encodes each state sequence $\boldsymbol{s}^{(t)}$ in parallel and produces hidden states $\boldsymbol{h}^{(t,0)}, \ldots, \boldsymbol{h}^{(t,M)}$. The second level LSTM takes as input the sequence of $\boldsymbol{h}^{(0,M)}, \ldots, \boldsymbol{h}^{(T,M)}$ and produces hidden state $\tilde{\boldsymbol{h}}^{(t)} \in \mathbb{R}^D$ and output state $\boldsymbol{o}^{(t)} \in \mathbb{R}^D$ for each time step. We predict the distribution over $c^{(t+1)}$ as $\text{softmax}(\boldsymbol{W}^{(out)}\boldsymbol{o}^{(t)})$, where $\boldsymbol{W}^{(out)} \in \mathbb{R}^{(|\mathcal{V}|+1) \times D}$. To predict $p_e^{(t+1)}$, we use a pointer network construction (Vinyals et al., 2015) where $\alpha^{(t,m)} = \langle \boldsymbol{o}^{(t)}, \boldsymbol{h}^{(t,m)} \rangle$ and $P(p_e^{(t+1)} = m) \propto \exp \alpha^{(t,m)}$. A useful elaboration is to provide a second path of context for the first level LSTM by concatenating the previous value of the token in the same position as an input. That is, we provide $(\boldsymbol{s}_m^{(t-1)}, \boldsymbol{s}_m^{(t)})$ as input at position $(t, m)$, letting $\boldsymbol{s}_m^{(-1)}$ be a special padding symbol. See Figure 3 (a) for an illustration.

**Baseline Implicit Model.** The natural application of the sequence-to-sequence framework is to consume the initial state $\boldsymbol{s}^{(0)}$ in the encoder and produce the sequence of $(p_i^{(t)}, c^{(t)})$ pairs in the decoder. The encoder is a standard LSTM. The decoder is slightly non-standard because each action is a pair. To deal with pairs as inputs, we concatenate an embedding of $p_i^{(t)}$ with an embedding of $c^{(t)}$. To produce pairs as outputs, we predict position and then content given position.

Formally, for position inputs, we embed each integer $m \in \{0, \ldots, M + T - 1\}$ by taking the $m^{th}$ column of a learned matrix $\boldsymbol{W}^{(\text{p-in})} \in \mathbb{R}^{D \times (M+T)}$. These are concatenated with content embeddings to produce inputs to the decoder, yielding hidden state $\boldsymbol{h}^{(t)}$ at step $t$ of the decoder. To predict position, we use a matrix $\boldsymbol{W}^{(\text{p-out})} \in \mathbb{R}^{(M+T) \times D}$ and define the distribution over position $p_i^{(t+1)}$ as $\text{softmax}(\boldsymbol{W}^{(\text{p-out})}\boldsymbol{h}^{(t)})$. Content is predicted by concatenating $\boldsymbol{h}^{(t)}$ with $\boldsymbol{W}_{:,p^{(t+1)}}^{(\text{p-in})}$, the embedding of the ground truth position, to get $\tilde{\boldsymbol{h}}^{(t)}$, and then the predicted distribution over content $c^{(t+1)}$ is $\text{softmax}(\boldsymbol{W}^{(\text{c-out})}\tilde{\boldsymbol{h}}^{(t)})$ where $\boldsymbol{W}^{(\text{c-out})} \in \mathbb{R}^{(|\mathcal{V}|+1) \times D}$ maps hidden states to content logits.

## 4 IMPLICIT ATTENTION MODEL

Here we develop a model that operates on the implicit representation but is better able to capture the sequence of the relationship of edit content to the context in which edits were made. The model is heavily inspired by Vaswani et al. (2017). At training time, the full sequence of edits is predicted in a single forward pass. There is an encoder that computes hidden representations of the initial state and all edits, then two decoder heads: the first decodes the position of each edit, and the second

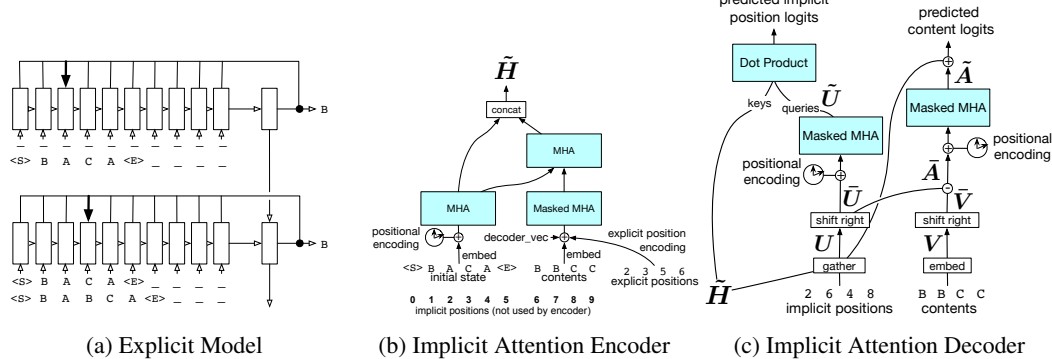

Figure 3: Diagrams of (a) baseline explicit model and (b, c) implicit attention model.

decodes the content of each edit given the position. The cross-time operations are implemented via the positional encoding and attention operations of Vaswani et al. (2017) with masking structure so that information cannot flow backward from future edits to past edits. Here, we give an overview of the model, focusing on the intuition and overall structure. An illustration appears in Figure 3 (b, c). Further details are in the Appendix, and we refer the reader to Vaswani et al. (2017) for full description of positional encodings, multi-head attention (MHA), and masked attention operations.

**Encode Initial State and Edits.** The first step is to encode the initial state $\boldsymbol{s} = (s_0, \ldots, s_M)$ and the edits $\boldsymbol{e} = ((p_i^{(1)}, c^{(1)}), \ldots, (p_i^{(T)}, c^{(T)}))$ into a matrix of hidden states $\tilde{\boldsymbol{H}} \in \mathbb{R}^{D \times (M+T)}$. We embed tokens and edits independently and then exchange information across the initial sequence and edits by MHA operations. After this step, we hope $\tilde{\boldsymbol{H}}$ includes all of the relevant context about each initial position and edit position. A diagram of the encoder structure appears in Figure 3 (b).

**Assemble the Chosen Contexts.** The decoder first gathers a matrix of 'contexts' $\boldsymbol{U} \in \mathbb{R}^{D \times T}$ where $\boldsymbol{U}_{:,t} = \tilde{\boldsymbol{H}}_{:,p_i^{(t)}}$. Intuitively, this step assembles hidden representations of the contexts in which previous edits were made into a new, more compact and more relevant subsequence.

**Predict the Next Positions.** The first head of the decoder looks for patterns in the sequence of columns of $\boldsymbol{U}$. It predicts a query vector $\tilde{\boldsymbol{u}}_t$ from $\boldsymbol{U}_{:,<t}$ that can be interpreted as a prediction of what context is expected to be edited next given the previous contexts that have been edited. The query vector is compared via inner product against each column of $\tilde{\boldsymbol{H}}$ as in Pointer networks (Vinyals et al., 2015) to get a probability distribution over next edit positions: $p(p_i^{(t)} = m) \propto \exp \tilde{\boldsymbol{u}}_t^\top \tilde{\boldsymbol{H}}_{:,m}$.

As an example, consider an edit sequence that appends B after each A in left-to-right order. If the encoder just preserves the identity and position of the corresponding content in $\tilde{\boldsymbol{H}}$, then the sequence of vectors in $\boldsymbol{U}$ will just be encodings of A with ascending positions. From this, it should be easy to learn to produce a $\tilde{\boldsymbol{u}}_t$ with content of A at a position beyond where the last edit was made.

**Predict the Next Contents.** The second head of the decoder predicts the content $c^{(t)}$ given all previous edits and the current position $p_i^{(t)}$. It first embeds the contents of all of the edits $(c^{(1)}, \ldots, c^{(T)})$ into a matrix of embeddings $\boldsymbol{V} \in \mathbb{R}^{D \times T}$. Let $\bar{\boldsymbol{V}}$ be a shifted version of $\boldsymbol{V}$; i.e., $\bar{\boldsymbol{V}}_{:,t} = \boldsymbol{V}_{:,t-1}$ with $\boldsymbol{V}_{:,-1} = \boldsymbol{0}$. In the *vanilla* decoder, we let $\boldsymbol{A} = \bar{\boldsymbol{V}} + \boldsymbol{U}$, which is the simplest way of combining information about previous content embeddings with current position embeddings. We pass $\boldsymbol{A}$ through a masked MHA operation to get $\tilde{\boldsymbol{A}}$. Each column $t$ of $\tilde{\boldsymbol{A}}$ is passed through a dense layer and softmax to get predicted logits for $c^{(t)}$.

The *analogical* decoder (see Figure 3 (c)) is motivated by the intuition that $\boldsymbol{V} - \boldsymbol{U}$ can be thought of as a matrix of analogies, because $\boldsymbol{V}$ encodes the content that was produced at each timestep, and $\boldsymbol{U}$ encodes the context. Intuitively, we might hope that $\boldsymbol{V}_{:,t}$ encodes "insert B" while the corresponding $\boldsymbol{U}_{:,t}$ encodes "there's a B and then an A," and the difference encodes "insert whatever comes before A". If so, then it might be easier to predict patterns in these analogies. To implement this, we let $\bar{\boldsymbol{U}}$ be a shifted version of $\boldsymbol{U}$ and $\bar{\boldsymbol{A}} = \bar{\boldsymbol{V}} - \bar{\boldsymbol{U}}$, then we predict the next analogies $\tilde{\boldsymbol{A}}$ using MHA operations over $\bar{\boldsymbol{A}}$. The predicted analogies are added to the *unshifted* current contexts $\tilde{\boldsymbol{A}} + \boldsymbol{U}$, which

is intuitively the predicted analogy applied to the context of the position where the current edit is about to be made. From this, we apply a dense layer and softmax to predict logits for $c^{(t)}$.

## 5 SYNTHETIC DATASETS

To study the ability of the various models to learn specific edit patterns and to isolate the ability to strongly generalize, we developed a suite of synthetic datasets based on regular expression replacements. The datasets are inspired by the kinds of edits we might see in real data, but they are simplified to allow clearer interpretation of results.

The datasets are based on generating a uniformly random initial string $s$ of length $L$ from a vocabulary $\mathcal{V}$ of size $V$. Each dataset is defined by a *pattern* and *replacement*. The pattern defines a criterion for matching a position in $s$ and the replacement defines what the pattern should be replaced with. Both pattern and replacement are sequences of characters from $\mathcal{V}$ (denoted by A, B, . . .) and *meta characters* (denoted by $x, y, z$), plus additional regular expression syntax: parentheses define groups in the pattern, and \N can be used in the replacement to refer to the sequence that was matched in the $N^{th}$ group. Meta characters will be replaced by different characters from $\mathcal{V}$ in each edit sequence. For example, let the pattern be '(.)$x$' and replacement be '\1$x$\1\1'. We might sample two initial strings BACA and DBBA, and sample $x$ to be replaced with A and B respectively. This would yield edit sequences with the following start and end states: BACA $\rightarrow$ BABBCACC and DBBA $\rightarrow$ DBDDBA.[1]

We define a *snapshot* to be the initial state and each intermediate state where a pattern has been replaced with a full replacement. We require that each synthetic instance be composed of at least four snapshots; otherwise the instance is rejected and we try again with a different random draw of initial state and meta character replacements. In the first example above, the snapshots would be [BACA, BABBCA, BABBCACC]. The full edit sequence is shown in Figure 2 (a). To create the edit sequences, we compute diffs between successive snapshots and apply the edits in the diff one token at a time from left to right. The set of edits that comprise a diff between two snapshots is not unique, and we use Python's built-in diff library to disambiguate between possible sets of edits. For each synthetic dataset, some number of edits need to be observed before the pattern becomes unambiguous. We call this the number of *conditioning steps* and give this many edits to the models before including their predictions in the total loss. We also create a "MultiTask" dataset that combines instances from all the above datasets, which is more challenging. A full list of the patterns and replacements that define the datasets appears in the Appendix.

## 6 EXPERIMENTS

The goal of the experiments is to understand the capabilities and limitations of the models discussed above, and to evaluate them on real data. Two main factors are how accurately the models can learn to recognize patterns in sequences of edits, and how well the models scale to large data. In our first set of experiments we study these questions in a simple setting; in the second set of experiments we evaluate on real data. In this section we evaluate three methods: the explicit model abbreviated as *E*, the implicit RNN model abbreviated as *IR*, and the implicit attention model from Section 4 with the analogical decoder abbreviated as *IA*. In the Appendix we evaluate variants of the implicit attention model that use the vanilla decoder and different update functions inside the MHA operations.

### 6.1 EXPERIMENTS ON SYNTHETIC DATA

**Accuracy.** The first question is how well the various models perform in terms of accuracy. For each of the synthetic tasks described in the Appendix we generated datasets of size 10k/1k/1k instances for train/dev/test, respectively. Initial sequences are of length $L = 30$, and the vocab size is $V = 10$. For all meta problems, we give the model conditional steps to let it recognize the meta character for each example. For evaluation, we measure average accuracy for each edit conditional on given past edits, where both position and content must be correct for an edit to be considered correct.

---

[1]The semantics are chosen to align with Python's re library. `import re; re.sub('(.)B', r'\1B \1\1', 'DBBA')` yields 'DBDDBA'.

Table 1: Test accuracies on synthetic datasets from step and hyperparameter setting with best dev accuracy. Results that are within .5% of the best accuracy are bolded. POMP: Position-Oracle Match-Pattern; E: Explicit baseline model; IR: Implicit baseline model; IA: Improved implicit model.

| | Non-Meta | | | | Meta | | | |
|---|---|---|---|---|---|---|---|---|
| | POMP | E | IR | IA | POMP | E | IR | IA |
| **Append1** | 100.0 | **100.0** | **100.0** | **99.9** | 100.0 | **99.9** | 13.9 | 83.0 |
| **ContextAppend11** | 100.0 | **100.0** | 98.6 | **99.9** | 100.0 | **99.5** | 2.5 | 96.3 |
| **ContextAppend13** | 100.0 | **100.0** | 98.6 | **100.0** | 100.0 | **100.0** | 73.5 | 98.9 |
| **Delete2** | 100.0 | **100.0** | **99.9** | **99.9** | 100.0 | **100.0** | 94.9 | **99.8** |
| **Flip11** | 100.0 | **99.7** | 82.8 | 98.8 | 99.9 | **99.1** | 10.0 | 92.4 |
| **Replace2** | 100.0 | **100.0** | **99.7** | **100.0** | 100.0 | **99.8** | 93.7 | 98.5 |
| **Surround11** | 100.0 | **100.0** | 91.2 | **99.8** | 99.9 | **99.6** | 12.1 | 98.5 |
| **ContextAppend31** | 99.9 | **99.5** | 76.9 | 98.5 | 95.9 | **95.7** | 18.0 | 94.3 |
| **ContextReverse31** | 99.9 | **99.4** | 72.6 | 98.1 | 95.9 | **95.3** | 14.4 | 94.4 |
| **ContextAppend33** | 99.7 | **99.6** | 76.3 | 98.9 | 95.9 | **99.3** | 73.3 | 97.5 |
| **ContextAppend52** | 37.6 | **99.2** | 74.6 | **99.3** | 11.9 | **97.2** | 63.0 | **97.5** |
| **ContextReverse51** | 37.6 | **99.0** | 59.5 | 95.2 | 11.9 | **94.7** | 22.6 | 92.4 |
| **Flip33** | 11.8 | **98.7** | 76.8 | **98.3** | 9.2 | **97.5** | 73.9 | 96.3 |
| **Surround33** | 11.8 | **99.6** | 79.5 | **99.6** | 9.2 | **99.1** | 74.9 | **99.0** |
| **MultiTask** | N/A | | | | - | 50.0 | 43.2 | **53.7** |

(a)   (b)   (c)   (d)

Figure 4: (a)-(c) Time required to process sequences during training, across $n$-gram problems with different numbers of insertions (10, 50, 100). Note that the y-axis scale changes across plots. (d) Token-level accuracy on the real dataset when limiting predictions to the contexts where the model is most confident. See text for more details.

To better understand how strong of generalization is required, we develop the *Position-Oracle Match-Pattern (POMP)* baseline. POMP assumes an oracle identifies the position where an edit needs to be made and marks the pattern part of the current state (using terminology from Section 5). Predictions for the changes needed to turn the pattern into the replacement are then done via pattern matching. If a test pattern appears anywhere in the training data, POMP is assumed to get all predictions correct; otherwise it guesses uniformly at random. We report the expected performance of POMP. In the cases where POMP achieves low accuracy, few of the patterns seen at test time appeared at training time, which shows that these tasks require a strong form of generalization. We can also interpret POMP as an upper bound on performance of any model based on counts of (pattern, replacement) pairs seen in training data, as would happen if we tried to adapt n-gram models to this task.

In Table 1 we report test performance for the hyperparameter setting and step that yield best dev performance. The explicit model and the improved implicit model can solve nearly all the tasks, even those that involve meta characters and relatively long sequences of replacements. Note that the POMP accuracy for many of these tasks is near chance-level performance, indicating that most test replacements were never seen at training time. In the Appendix, we provide more statistics about the synthetic datasets that give additional explanation for the varying performance across tasks.

**Evaluating Scalability.** Here we explore questions of scalability as the length of the state and the number of edits grow. We use a simple dataset where the pattern is a single character and the replacement comes from a randomly sampled $n$-gram language model. In all cases we use $n = 3$ and set the number of insertions to one of $\{10, 50, 100\}$. Note that this simultaneously increases the size of the explicit state ($M$) and the number of edits ($T$). The scalability metric is the average time required to run training on 128 sequences on a single P100 GPU, averaged over 1000 training steps.

As shown in Figure 4, the explicit model is consistently more expensive than the implicit models, and the gap grows as the size of the data increases. The length-100 insertion sequences are ten times smaller than the sequences in the real dataset, but already there is an order of magnitude difference in runtime. The attention models generally take 50% to 75% of the time of the implicit RNN models.

## 6.2 Experiments on Real Data

We have obtained a large-scale dataset of code edits in Python. Each time a developer saved a file, a snapshot was recorded, making the dataset much more fine-grained than other collection methods like Git commits. First, we converted source code into tokens using Python's built-in tokenization library and then converted tokens into subword tokens using the subword tokenizer of Vaswani et al. (2017) with a target subword vocabulary size of 4096. This yields sequences of snapshots of subword tokens that we process into sequences of edits as described in Section 5. In total the dataset includes 8 million edits from 5700 software developers. We set the number of conditioning steps to 0 for real data. We grouped snapshots into instances involving about 100 edits each, and we pruned instances that included states with more than 1k subword tokens. We divided instances randomly into train, dev, and test with proportions 80%/10%/10%.

For this experiment, we evaluate the performance of the E, IR, and IA models. The subtoken-level test accuracies are 51% for the explicit baseline (E), 55.5% for the implicit baseline (IR), and 61.1% for the improved implicit model (IA). These numbers are calculated using parameters from the step and hyperparameter setting that achieved the best subtoken-level accuracy on the dev set. Our main take-away from these experiments is that the IA model provides a good tradeoff. It achieves the best accuracy, and it had less issue with memory constraints than the explicit model. As we move towards training on larger datasets with longer initial states, we are optimistic about building on the IA model.

Finally, we evaluate the models on their ability to auto-regressively predict a sequence of subtokens up to the next token boundary. We are particularly interested in the setting where models only make predictions when they are confident, which could be important for usability of an eventual edit suggestion system. To decode we use a greedy decoding strategy of generating the most likely next subtoken at each step until reaching the end of a token. As a confidence measure, we use the log probability assigned to the sequence of subtoken predictions.

In Figure 4d, we sort predictions based upon their confidence, and then for each possible confidence threshold we report results. The x-axis denotes the percentile of confidence scores on which we make predictions (so at confidence percentile 75%, we make predictions on the 25% of instances where the model is most confident), and the y-axis shows the average accuracy amongst the instances where a prediction is made. The IA model outperforms the other models across confidence thresholds and when the model is confident, accuracy is correspondingly high. This suggests that the model's confidence could be useful for deciding when to trigger suggestions in, e.g., an edit suggestion tool.

## 7 Related Work

**Sequence to Sequence and Attention Models.** Modeling sequences of edits could be approached via minor modifications to the sequence-to-sequence paradigm (Sutskever et al., 2014) as in our implicit baseline. For explicit data, we can apply hierarchical recurrent neural network-like models (Serban et al., 2016) to encode sequences of sequences. The baseline explicit model is an elaboration on this idea.

Attention mechanisms (Bahdanau et al., 2015) are widely used for machine translation (Bahdanau et al., 2015), summarization (Rush et al., 2016), and recognizing textual entailment (Rocktäschel et al., 2016). Recently, Vaswani et al. (2017) showed that a combination of attention and positional encodings can obviate the need for standard recurrent connections in autoregressive neural networks. Pointer networks (Vinyals et al., 2015) use an attention-like mechanism to select positions in an input sequence. These are important components for the model in Section 4.

**Generation by Iterative Refinement.** There are several works that generate structured objects via *iterative refinement*. The structured objects could be images (Gregor et al., 2015; Denton et al., 2015; Karras et al., 2018), translations (Li et al., 2017; Novak et al., 2016; Niehues et al., 2016; Lee et al., 2018), or source code (Gupta et al., 2017; Shin et al., 2018). These methods produce "draft" objects

that are refined via one or more steps of edits. Of these, the most similar to our work is DeepFix (Gupta et al., 2017), which iteratively predicts line numbers and replacement lines to fix errors in student programming assignments. However, like other iterative refinement methods, the refinement is aimed towards a pre-defined end goal (fix all the errors), and thus is not concerned with deriving intent from the past edits. In DeepFix, for example, the only information about past edits that is propagated forward is the resulting state after applying the edits, which will not generally contain enough information to disambiguate the intent of the edit sequence. Schmaltz et al. (2017) studies the problem of grammar correction. While the use of diff-like concepts is superficially similar, like the above, the problem is phrased as mapping from an incorrect source sentence to a corrected target sentence. There is no concept of extracting intent from earlier edits in order to predict future edits.

To emphasize the difference, note that all the approaches would fail on our Meta problems, because the goal only reveals itself after some edits are observed. Any task or method that has only initial and final states would suffer from the same issues.

**Software Engineering and Document Editing.** Some work in software engineering has modeled edits to code, but it operates at a coarser level of granularity. For example, Ying et al. (2004); Zimmermann et al. (2005); Hu et al. (2010) look at development histories and identify files that are commonly changed together. Our work is more fine-grained and considers the temporal sequence of edits. There is a large body of work on statistical models of source code (see Allamanis et al. (2017)). There are source code models based on $n$-grams (Hindle et al., 2012; Allamanis & Sutton, 2013), grammars (Allamanis & Sutton, 2014), neural networks (Raychev et al., 2014; White et al., 2015; Ling et al., 2016; Bhoopchand et al., 2016), and combinations of the two (Maddison & Tarlow, 2014; Yin & Neubig, 2017). Other notable models include Bielik et al. (2016), Raychev et al. (2016), and Hellendoorn & Devanbu (2017). All of these model a single snapshot of code.

Our formulation, in contrast, models the process of constructing code as it happens in the real world. The benefit is there are patterns available in edit histories that are not be present in a static snapshot, and the models are trained to make predictions in realistic contexts. A limitation of our work is that we treat code as a sequence of tokens rather than as tree-structured data with rich semantics like some of the above. However, source code representation is mostly orthogonal to the edits-based formulation. In future work it would be worth exploring edits to tree- or graph-based code representations.

There are few works we know of that infers intent from a history of edits. One such example is Raza et al. (2014). Given past edits to a presentation document, the task is to predict the analogous next edit. A major difference is that they create a high-level domain specific language for edits, and predictions are made with a non-learned heuristic. In contrast, we define a general space of possible edits and learn the patterns, which is better suited to noisy real-world settings. Other such works include Nguyen et al. (2016) and Zimmermann et al. (2005). In comparison to our work, which tries to infer both position and content from full edit histories, Nguyen et al. (2016) predicts only the content, in the form of a method call from an API, given the location. Zimmermann et al. (2005) instead predicts the location of the next change, given the location of the previous change. Both approaches rely on frequency counts to determine correlations between edit pairs in order to produce a ranking, and work at a coarser granularity than tokens. In concurrent work, Paletov et al. (2018) studies the problem of deriving rules for usage of cryptography APIs from changes in code, which is similar in spirit to our work in trying to derive intent from a history of changes to code.

## 8 Discussion

In this work, we have formulated the problem of learning from past edits in order to predict future edits, developed models of edit sequences that are capable of strong generalization, and demonstrated the applicability of the formulation to large-scale source code edits data.

An unrealistic assumption that we have made is that the edits between snapshots are performed in left-to-right order. An alternative formulation that could be worth exploring is to frame this as learning from weak supervision. One could imagine a formulation where the order of edits between snapshots is a latent variable that must be inferred during the learning process.

There are a variety of possible application extensions. In the context of developer tools, we are particularly interested in conditioning on past edits to make other kinds of predictions. For example, we could also condition on a cursor position and study how edit histories can be used to improve

traditional autocomplete systems that ignore edit histories. Another example is predicting what code search queries a developer will issue next given their recent edits. In general, there are many things that we may want to predict about what a developer will do next. We believe edit histories contain significant useful information, and the formulation and models proposed in this work are a good starting point for learning to use this information.

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

# Appendix

OVERVIEW

- Section A gives more information on the synthetic datasets.
- Section B gives additional details for the synthetic data experiments.
- Section C gives additional details for the real data experiments.
- Section D gives a more detailed description of the IA models and its variants.
- Section E gives more details on the variant of multihead attention used in the IA models.
- Section F provides additional experiments on the IA model variants.

## A  ADDITIONAL SYNTHETIC DATASET INFORMATION

The regular expression patterns and replacements that define the synthetic datasets are listed in Table 2. In this table we also show additional computed properties of the synthetic datasets.

The number of edits per replacement (EPR) helps to explain why some problems are harder than others. Problems that only repeat their context-based action a single time, rather than multiple times, are harder and have correspondingly lower accuracies. For example, MetaContextAppend11 is harder than MetaContextAppend13. This is because accuracy is measured in a setting where the model conditions on ground truth for past edits, so previous correct instances of the action are available for predicting future instances of the action. This is visible in the higher accuracies for those problems with multiple wildcards in their pattern and higher EPR, corresponding to multiple actions.

A higher average context displacement (ACD) also is indicative of harder problems, and correlates with lower accuracies from models E, IA-ag, and IR. ACD measures the distance between characters in the regex patterns and where they are used in the replacement.

We also observe a moderate correlation between accuracy of E, IA-ag, and IR and the accuracy of POMP on the non-meta problems. This suggests the problems where fewer of the test time patterns appeared at training time are harder than those where more of the test time patterns appeared at training time. The IR model in particular tends to have lower accuracy for the problems that have more novel patterns at test time.

| Dataset | Pattern | Replacement | EPR | ACD | POMP | E | IA-ag | IR |
|---|---|---|---|---|---|---|---|---|
| Append1 | A | AB | 1 | 0 | 100.0 | 100.0 | 99.9 | 100.0 |
| ContextAppend11 | (.)A | \1A\1 | 1 | 2 | 100.0 | 100.0 | 99.9 | 98.6 |
| ContextAppend13 | (.)A | \1A\1\1\1 | 3 | 3 | 100.0 | 100.0 | 100.0 | 98.6 |
| ContextAppend31 | (...)A | \1A\1 | 3 | 4 | 100.0 | 99.5 | 98.5 | 76.9 |
| ContextAppend33 | (...)A | \1A\1\1\1 | 9 | 7 | 99.7 | 99.6 | 98.9 | 76.3 |
| ContextAppend52 | (.....)A | \1A\1\1 | 10 | 8.5 | 37.6 | 99.2 | 99.3 | 74.6 |
| ContextReverse31 | (.)(.)(.)A | \1\2\3A\3\2\1 | 3 | 4 | 100.0 | 99.4 | 98.1 | 72.6 |
| ContextReverse51 | (.)(.)(.)(.)(.)A | \1\2\3\4\5A\5\4\3\2\1 | 5 | 6 | 37.6 | 99.0 | 95.2 | 59.5 |
| Delete2 | AA |  | 2 | 0 | 100.0 | 100.0 | 99.9 | 99.9 |
| Flip11 | (.)A(.) | \2\1A\2\1 | 2 | 3 | 100.0 | 99.7 | 98.8 | 82.8 |
| Flip33 | (...)A(...) | \2\2\2\1A\2\1\1\1 | 18 | 10 | 11.8 | 98.7 | 98.3 | 76.8 |
| Replace2 | AA | BB | 4 | 0 | 100.0 | 100.0 | 100.0 | 99.7 |
| Surround11 | (.)A(.) | \1\1A\2\2 | 2 | 1 | 100.0 | 100.0 | 99.8 | 91.2 |
| Surround33 | (...)A(...) | \1\1\1\1A\2\2\2\2 | 18 | 6 | 11.8 | 99.6 | 99.6 | 79.5 |
| MetaAppend1 | x | xy | 1 | 0 | 100.0 | 99.9 | 83.0 | 13.9 |
| MetaContextAppend11 | (.)x | \1x\1 | 1 | 2 | 100.0 | 99.5 | 96.3 | 2.5 |
| MetaContextAppend13 | (.)x | \1x\1\1\1 | 3 | 3 | 100.0 | 100.0 | 98.9 | 73.5 |
| MetaContextAppend31 | (...)x | \1x\1 | 3 | 4 | 95.9 | 95.7 | 94.3 | 18.0 |
| MetaContextAppend33 | (...)x | \1x\1\1\1 | 9 | 7 | 95.9 | 99.3 | 97.5 | 73.3 |
| MetaContextAppend52 | (.....)x | \1x\1\1 | 10 | 8.5 | 11.9 | 97.2 | 97.5 | 63.0 |
| MetaContextReverse31 | (.)(.)(.)x | \1\2\3x\3\2\1 | 3 | 4 | 95.9 | 95.3 | 94.4 | 14.4 |
| MetaContextReverse51 | (.)(.)(.)(.)(.)x | \1\2\3\4\5x\5\4\3\2\1 | 5 | 6 | 11.9 | 94.7 | 92.4 | 22.6 |
| MetaDelete2 | xx |  | 2 | 0 | 100.0 | 100.0 | 99.8 | 94.9 |
| MetaFlip11 | (.)x(.) | \2\1x\2\1 | 2 | 3 | 100.0 | 99.1 | 92.4 | 10.0 |
| MetaFlip33 | (...)x(...) | \2\2\2\1x\2\1\1\1 | 18 | 10 | 9.2 | 97.5 | 96.3 | 73.9 |
| MetaReplace2 | xx | yy | 4 | 0 | 100.0 | 99.8 | 98.5 | 93.7 |
| MetaSurround11 | (.)x(.) | \1\1x\2\2 | 2 | 1 | 100.0 | 99.6 | 98.5 | 12.1 |
| MetaSurround33 | (...)x(...) | \1\1\1\1x\2\2\2\2 | 18 | 6 | 9.2 | 99.1 | 99.0 | 74.9 |

Table 2: Regular expressions used to generate the synthetic datasets, and properties of the synthetic datasets. Edits per replacement (EPR) is the number of edits required for a typical replacement of the pattern with the replacement. Average context displacement (ACD) measures the average distance between a regex capture group in the pattern and where the characters in that capture group appear in the replacement.

**MetaSurround33**

Meta Character: F

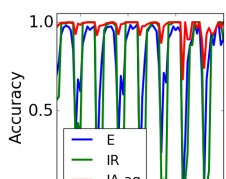

**MetaContextAppend31**

Meta Character: F

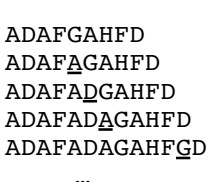

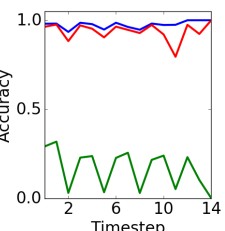

Figure 5: Zooming in on performance of illustrative synthetic problems. Left: Two example edit sequences (shortened for illustrative purposes). Each row is a timestep, and the underlined character marks where the insertion was made. Right: Dev accuracy vs timestep of prediction averaged over the dev set.

## B  SYNTHETIC DATA EXPERIMENTS

### B.1  ADDITIONAL EXPERIMENT DETAILS

For the synthetic dataset experiments, for each method, we performed a grid search over learning rate (.005, .001, .0005, .0001) and hidden size (128, 256, 512). We use the ADAM optimization algorithm with default Tensorflow settings for other parameters. We clip gradients at 1.0 by global gradients norm.

### B.2  ADDITIONAL SCALABILITY EXPERIMENT DETAILS

We test all models on hidden sizes (128, 256, 512) with batch size 64. The metric we use for scalability is the average time required to run training on 128 sequences on a single P100 GPU, averaged over 1000 training steps. Under this metric, the explicit model pays an additional cost for its high memory usage. On the larger datasets (and in the code edits dataset), it is not possible to fit a batch size of 64 in memory for hidden size 512, so we need to run several training steps with batch size 32. While this may appear to unfairly penalize the explicit model, this is the tradeoff that we had to face when running experiments on the real data, so we believe it to be an accurate reflection of the practical ability to scale up each type of model.

### B.3  ANALYZING ERRORS

In Figure 5 we examine the performance of two illustrative synthetic problems at each of the timesteps of prediction. The plots show combined position and content accuracies averaged over validation examples. For all models, this accuracy dips when the target position changes non-trivially between consecutive timesteps, which occurs when a pattern is complete and the model must decide where to edit next. This shows the cross-pattern jumps are the most challenging predictions. For the best performing models, the accuracy stays near 1 when the position merely increments by 1 between timesteps, and decreases for the cross-pattern jumps. In these examples the substitution pattern is of fixed size, so there is regularity on when the cross-pattern jumps happen, and hence also regularity in the dips.

## C  REAL DATA EXPERIMENTS

### C.1  ADDITIONAL EXPERIMENT details

In this experiment, we evaluate the performance of the E, IR, and IA models on the code edits dataset. For each of these models, we performed a grid search over the hyperparameter space. We evaluate the learning rates .005, .001, .0005, and .0001. For the implicit models, we evaluate the hidden sizes 128, 256, and 512, and batch size 64. For the explicit models, we evaluate hidden sizes 32 and 64 (with batch size 10), 128 (with batch size 5), and 256 and 512 (with batch size 1). We decrease the batch size considered as we increase the hidden size so that the model fits in memory. We trained each model variant on a single P100 GPU for 48 hours.

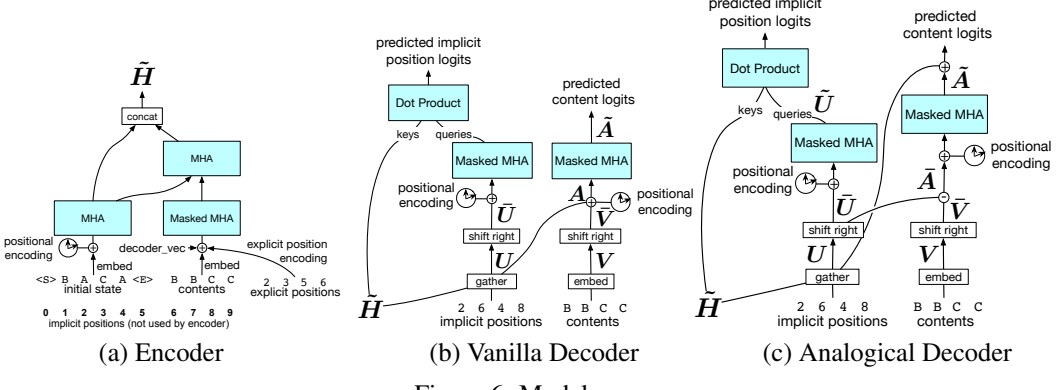

(a) Encoder      (b) Vanilla Decoder      (c) Analogical Decoder

Figure 6: Models.

# D  IMPLICIT ATTENTIONAL MODEL - LONGER DESCRIPTION

## D.1  ENCODER

In contrast to before, here the task of the encoder is to convert the $M$ initial state tokens *and* the $T$ edits into a matrix of hidden states $\tilde{\boldsymbol{H}} \in \mathbb{R}^{D \times (M+T)}$. The first $M$ columns represent the context around each position in the initial state, and these embeddings must only depend on $\boldsymbol{s}^{(0)}$. The last $T$ columns represent the context around each token produced by the edit sequence $\boldsymbol{e}$. The $t^{th}$ of these columns can only depend on $\boldsymbol{s}^{(0)}$ and edits $e_1, \ldots, e_t$.

One design constraint imposed by Vaswani et al. (2017) is that there should be no sequential dependence in the encoder, and we would like to impose the same constraint on our models to aid in scalability. That is, given embeddings of each of the inputs and edits, the hidden states for each of the inputs and edits can be computed in parallel. However, this raises a challenge of how to compute a positional embedding of the tokens generated by edits. Suppose we have an initial sequence ABC and then insert XY after A. How should we encode the position of X and Y such that attention operations in the encoder can produce a hidden representation of Y encoding the information that Y comes one position after X and two positions after A?

Our approach to this problem is to compute the embedding of an edit as a sum of three terms: an embedding of the content, a positional encoding of the *explicit* position, and a shared learned embedding to represent that the content was produced by an edit (as opposed to being present in the initial state). Note that using the explicit position means that multiple tokens may have the same position; however, the justification is that relative positions still make sense, and the third component of the sum can be used to disambiguate between positions in the initial state and those produced by edits. From the embeddings of initial state and edit sequence, we apply a sequence of MHA operations illustrated in Figure 6 (a) to produce the encoder output $\tilde{\boldsymbol{H}}$.

## D.2  DECODER

Given $\tilde{\boldsymbol{H}}$, we can use the implicit position indexing illustrated in Figure 6 (b) to reference the hidden state at the position of edit $t$ as $\tilde{\boldsymbol{H}}_{:,p_i^{(t)}}$. Thus we can assemble a matrix of "contexts" $\boldsymbol{U} \in \mathbb{R}^{D \times T}$ where $\boldsymbol{U}_{:,t} = \tilde{\boldsymbol{H}}_{:,p_i^{(t)}}$ for each $t$. We can also assemble a matrix of "contents" of each edit $\boldsymbol{V} \in \mathbb{R}^{D \times T}$, where $\boldsymbol{V}_{:,t}$ is an embedding of $c^{(t)}$. Intuitively, we hope the context vectors to capture the relevant context around the position that was chosen (e.g., it is after a B), while the content vectors represent the content of the edit (e.g., in that context, insert a B).

To predict position, we first shift $\boldsymbol{U}$ forward in time to get $\bar{\boldsymbol{U}}$ where $\bar{\boldsymbol{U}}_{:,t} = \boldsymbol{U}_{:,t-1}$ (letting $\boldsymbol{U}_{:,-1} = \boldsymbol{0}$). We then apply a masked MHA operation to get $\tilde{\boldsymbol{U}} = \text{MHA}(\bar{\boldsymbol{U}})$. The result is used as keys for a pointer network-like construction for predicting position. Specifically, we let $\alpha_{t,m} = \left\langle \tilde{\boldsymbol{U}}_{:,t}, \tilde{\boldsymbol{H}}_{:,m} \right\rangle$ be the compatibility between the predicted context $\tilde{\boldsymbol{U}}_{:,t}$ and the $m^{th}$ context in $\tilde{\boldsymbol{H}}$, and then define

the predicted distribution for position to be $P(p_i^{(t)} = m) = \frac{\exp \alpha_{t,m}}{\sum_{m'} \exp \alpha_{t,m'}}$. See the left column of Figure 6 (b) and (c) for an illustration of the position prediction component of the decoder.

To predict content $c^{(t)}$, we also condition on $p_i^{(t)}$ because of our chosen convention to predict position and then content given position. We consider two content decoders. The "Vanilla" decoder (see Figure 6 (b)) takes the un-shifted contexts matrix $U$ used in the position decoder. Recall that the $t^{th}$ column has an encoding of the context where the $t^{th}$ edit was made. This matrix is added to a shifted version of the content matrix $\bar{V}$ (i.e., $\bar{V}_{:,t} = V_{:,t-1}$). Thus, the $t^{th}$ column of the result has information about the content of the previous edit and the context of the current edit. This summed matrix is passed through a separate masked MHA module that passes information only forward in time to yield $\tilde{V}$, which incorporates information about previous contents and previous plus current contexts. From this, we predict the distribution over $c^{(t)}$ as $\mathrm{softmax}(W^{(\text{c-out})}\tilde{V}_{:,t})$.

The "Analogical" decoder (see Figure 6 (c)) is motivated by the intuition that $V - U$ can be thought of as a matrix of analogies, because $V$ encodes the content that was produced at each timestep, and $U$ encodes the context. We might hope that $V_{:,t}$ encodes "insert B" while the corresponding $U_{:,t}$ encodes "there's a B and then an A," and the difference encodes "insert whatever comes before A". If so, then it might be easier to predict patterns in these analogies. To implement this, we construct previous analogies $\bar{A} = \bar{V} - \bar{U}$, then we predict the next analogies as $\tilde{A} = \mathrm{MHA}(\bar{A})$. The next analogies are added to the *unshifted* current contexts $\tilde{A} + U$, which is intuitively the predicted analogy applied to the context of the position where the current edit is about to be made. From this, we apply a dense layer and softmax to predict the distribution over $c^{(t)}$.

High level pseudocode is provided below. `s2s_mha` denotes exchanging information between tokens within the initial state; `s2e_mha` denotes passing information from tokens in the initial state to edits; `e2e_mha` denotes exchanging information between edits and applies masking to avoid passing information backward in time.

```
# Encoder
state_embeddings = embed_tokens(initial_state_tokens)
state_hiddens = s2s_mha(state_embeddings)

edit_embeddings = embed_edits(edit_positions, edit_contents)
edit_hiddens = e2e_mha(edit_embeddings)
edit_hiddens = s2e_mha(state_hiddens, edit_hiddens)

H = concat(state_hiddens, edit_hiddens)

# Decoder
U = gather(H, target_positions)

# Predict position
prev_U = shift_forward_in_time(U) + timing_signal(U)
predicted_U = e2e_mha(prev_U)
position_logits = pointer_probs(query=predicted_U, keys=H)

# Predict content
V = embed_tokens(target_contents)
if vanilla_decoder:
    prev_V = shift_forward_in_time(V) + timing_signal(V)
    predicted_V = e2e_mha(U + prev_V)
else:
    prev_V = shift_forward_in_time(V) + timing_signal(V)
    predicted_V = e2e_mha(prev_V - prev_U) + U
```

## E    ADDITIONAL MULTIHEAD ATTENTION MODULE DETAILS

In its most general form, a multihead attention (MHA) module described in the main text takes as input three matrices:

- A keys matrix $K \in \mathbb{R}^{D \times M}$,
- A values matrix $V \in \mathbb{R}^{D \times M}$, and
- A queries matrix $Q \in \mathbb{R}^{D \times N}$.

If only one matrix is provided, then it is used as $K$, $V$, and $Q$. If two matrices are provided, the first is used as $Q$, and the second is used as $K$ and $V$.

Deviating slightly from Vaswani et al. (2017) by grouping together surrounding operations, the module is composed of the following operations:

- Add positional encoding to $Q$,
- Apply attention to get a result matrix $R \in \mathbb{R}^{D \times N}$,
- Apply an aggregation operation $\text{Agg}(Q, R)$ and return the result.

The positional encoding is the same as in Vaswani et al. (2017); we add their sinusoidal positional embedding of integer $n$ to the $n^{th}$ column of $Q$. The attention operation is the multihead attention operation as described in Vaswani et al. (2017), where we use 8 heads throughout. The aggregation operation is either a simple sum $Q + R$ or a GRU operation. That is, we treat $Q$ as a matrix of previous hidden states and $R$ as inputs, and then we apply the update that a GRU would use to compute the current hidden states. The potential benefit of this construction is that there is a learned gating function that can decide how much to use $Q$ and how much to use $R$ in the final output of the module in an instance-dependent way.

When we use GRU aggregation, we append a $g$ to the method name, and when we use a sum aggregation we do not append a character. Paired with the Vanilla ($v$) and Analogical ($a$) decoders described in the main text, this describes all the implicit model variants:

- IA-v: vanilla decoder, sum aggregation,
- IA-vg: vanilla decoder, GRU aggregation,
- IA-a: analogical decoder, sum aggregation,
- IA-ag: analogical decoder, GRU aggregation,

## F    ADDITIONAL IMPROVED IMPLICIT MODEL EXPERIMENTS

We repeated the synthetic experiments from the main paper but on all the implicit attentional model variants. Results appear in Table 3.

| | IA-vg | IA-a | IA-v | IA-ag |
|---|---|---|---|---|
| **Append1** | **99.9** | **99.9** | **100.0** | **99.9** |
| **Append3** | **100.0** | **99.9** | **100.0** | **100.0** |
| **ContextAppend11** | **99.9** | **99.5** | **99.5** | **99.9** |
| **ContextAppend13** | **99.9** | **99.8** | **99.8** | **100.0** |
| **ContextAppend31** | **98.0** | **98.5** | **98.3** | **98.5** |
| **ContextAppend33** | **99.2** | **98.9** | **98.9** | **98.9** |
| **ContextAppend52** | **99.2** | **99.1** | **99.0** | **99.3** |
| **ContextReverse31** | **97.9** | 97.5 | 97.3 | **98.1** |
| **ContextReverse51** | **96.5** | **96.1** | **96.5** | 95.2 |
| **Delete2** | **99.9** | **99.9** | **99.9** | **99.9** |
| **Flip11** | **99.3** | **98.8** | **98.9** | **98.8** |
| **Flip33** | **98.8** | 98.2 | **98.9** | 98.3 |
| **Replace2** | **100.0** | **99.9** | **100.0** | **100.0** |
| **Surround11** | **99.8** | **99.8** | **99.7** | **99.8** |
| **Surround33** | **99.6** | **99.5** | **99.5** | **99.6** |
| **MetaAppend1** | 84.1 | **84.7** | 83.4 | 83.0 |
| **MetaContextAppend11** | **95.8** | 95.2 | 92.8 | **96.3** |
| **MetaContextAppend13** | **99.1** | **98.6** | 98.3 | **98.9** |
| **MetaContextAppend31** | 93.1 | 91.6 | 92.5 | **94.3** |
| **MetaContextAppend33** | **98.1** | 97.7 | **98.1** | 97.5 |
| **MetaContextAppend52** | **97.5** | **97.8** | **97.7** | **97.5** |
| **MetaContextReverse31** | 93.0 | 91.1 | 92.4 | **94.4** |
| **MetaContextReverse51** | **93.0** | 92.2 | 91.6 | 92.4 |
| **MetaDelete2** | **99.5** | **99.4** | **99.5** | **99.8** |
| **MetaFlip11** | 48.1 | 85.8 | 86.9 | **92.4** |
| **MetaFlip33** | 96.9 | **97.6** | **97.4** | 96.3 |
| **MetaReplace2** | 98.6 | 98.3 | **99.7** | 98.5 |
| **MetaSurround11** | **98.1** | 96.9 | 96.5 | **98.5** |
| **MetaSurround33** | **99.1** | **99.1** | **99.1** | 99.0 |

Table 3: Comparing variations of the Improved Implicit models. Table reports test accuracy at hyperparameter and step that achieved best dev accuracy. IA-v: Vanilla decoder with sum aggregator; IA-vg: Vanilla decoder with GRU aggregator; IA-a: Analogy decoder with sum aggregator; IA-ag: Analogy decoder with GRU aggregator.

