# OpenReview forum: "Neural Networks for Modeling Source Code Edits"
_ICLR.cc/2019/Conference_

### Official Review · AnonReviewer1 · 2018-11-02
**Solid step forward for NNs and source code**

**Rating:** 6
**Confidence:** 4

**Review:**

Summary: The authors study building models for edits in source code. The application is obvious: a system to accurately predict what the next edit should be would be very valuable for developers. Here, edits are modeled by two types of sequences: one that tracks the state of all edits at each time step (and is thus very long), and one that contains the initial step and a changelist that contains the minimal information required to derive the state at any time. The authors train models on top of both of these representations, with the idea being to match the performance of the explicit (heavy) model with the implicit model. This is shown to be challenging, but a clever model is introduced that achieves this, and is thus the best of both worlds. There are synthetic and real-world code (text) edit experiments.

Strengths: The problem is well-posed and well-motivated. There's a nice application of powerful existing models, combined and tailored to the current work. The writing is generally quite clear. The number of experiments is quite solid.

Weaknesses: The main flaw is that nothing here is really specifically for souce code; the authors are really just modeling edits in text sequences. There's not an obvious way to integrate the kinds of constraints that source code typically satisfies either. There's some confusion (for me) about the implicit/explicit representations. More questions below.

Verdict: This is a pretty solid paper. It doesn't quite match up to its title, but it sets out a clearly defined problem, achieves its aims, and introduces some nice tricks. Although it doesn't produce anything genuinely groundbreaking, it seems like a nice step forward.

Comments and Questions:

- The problem is written in the context of source code, but it's really setup just for text sequences, which is a broader problem. Is there a way the authors take can advantage of the structural requirements for source code? I don't see an obvious way, but I'm curious what the authors think.

- What's the benefit of using the implicit representation for the positions? The explicit/implicit  position forms are basically just using the permutation or the inverse permutation form, which are equivalent. I don't see directly what's saved here, the alphabet size and the number of integers to store is the same.

- Similar question. The implicit likelihood is s^0, e^(1),...,e^(t-1), with the e^(i)'s being based on the implicit representations of the positions. Seems like you could do this with the *explicit* positions just fine, they carry enough information to derive s^(i) from s^(i-1). That is, the explicit/implicit problems are not really related to the explicit/implicit position representations.

- Just wanted to point out that this type of approach to sequences and edits has been studied pretty often in the information/coding theory communities, especially in the area of synchronization. There, the idea is to create the minimal "changelist" of insertions/deletions from two versions of a file. This could come in handy when building the datasets. See, for example, Sala et al "Synchronizing files from a large number of insertions and deletions".

- The problem statement should be stated a bit more rigorously. We'd like to say that the initial state is drawn from some distribution and that the state at each time forms a stochastic process with some transition law. As it stands the problem isn't well-defined, since with no probability distribution, there's nothing to predict and no likelihood.

- The "analogical decoder" idea is really nice.

- For the synthetic dataset, why are you selecting a random initial string, rather than using some existing generative text or source code model, which would get you synthetic data that more closely resembles code?

- I really liked the idea of using an oracle that gives the position as upper bound. Would it make sense to also have the opposite oracle that gives the edit symbol, but doesn't tell the location? I'm really curious which is the "harder" task, predicting the next symbol or the next location. In the information-theory setting, these two are actually equally hard, but the real-world setting might be pretty different. It would also be interesting to train models on top of the POMP. That would produce genuine upper bounds to the model performances.

- The explicit baseline model performs very well on all the edit types in Table 1. Are there cases where even this explicit case works poorly? Is the improved implicit model *always* upper bounded by the explicit model (to me it seems like the answer should always be yes, but it would be interesting to check it out for cases where explicit is not very high accuracy).

---

> ### Author Response · Authors · 2018-11-26
> **Response to “Solid step forward for NNs and source code”**
>
> Thanks for the positive review and the careful attention to detail! Questions are answered inline below.
>
> > Is there a way the authors take can advantage of the structural requirements for source code? I don't see an obvious way, but I'm curious what the authors think.
>
> This is an excellent insight, and we have definitely considered extensions of our models to structured data. As this was a question that was highlighted by multiple reviewers, please see our response in the general comment to all reviewers. The short answer is that we do believe such extensions are possible, but in this work we wanted to stay focused on formulating the problem itself before taking advantage of more domain-specific structures.
>
> > What's the benefit of using the implicit representation for the positions? The explicit/implicit position forms are basically just using the permutation or the inverse permutation form, which are equivalent. I don't see directly what's saved here, the alphabet size and the number of integers to store is the same.
>
> A key property of implicit positions is that they don’t change as more edits are made. This enables the training efficiency like in Vaswani et al, where we can train on all output steps with a single forward-backward pass. We wouldn’t be able to do this with explicit positions that potentially changed after each edit.
>
> > Similar question. The implicit likelihood is s^0, e^(1),...,e^(t-1), with the e^(i)'s being based on the implicit representations of the positions. Seems like you could do this with the *explicit* positions just fine, they carry enough information to derive s^(i) from s^(i-1). That is, the explicit/implicit problems are not really related to the explicit/implicit position representations.
>
> This is fair. It’s possible to derive the explicit positions given the previous edits in the implicit representation (like applying a diff), and indeed, in the experiments we are leveraging the fact that it is valid to compare results between models trained on the different representations. The point we’re trying to make here is just one of how the data is presented to the neural network. In the implicit case, it’s up to the neural network to “learn how to apply the diff” whereas in the explicit case, an external deterministic algorithm applies the diff for the neural network. So implicit vs explicit is more about distinguishing between what kinds of representations the network is given vs being forced to learn. We’ll clarify the wording in “Problem Statement” to say that there is only one problem, which is to predict next edits given initial state and previous edits, but two families of architectures: explicit assumes an external algorithm applies the diffs to get explicit states that are fed into the network, and implicit works directly off the edits.
>
> > For the synthetic dataset, why are you selecting a random initial string, rather than using some existing generative text or source code model, which would get you synthetic data that more closely resembles code?
>
> It would certainly be possible. The random initial strings were simpler because we have control over the vocabulary size and length of sequences, which allowed us to control how long the initial strings were and how many edits were applied. With real code, we think it would introduce a few more confounding factors.
>
> > I really liked the idea of using an oracle that gives the position as upper bound. Would it make sense to also have the opposite oracle that gives the edit symbol, but doesn't tell the location? I'm really curious which is the "harder" task, predicting the next symbol or the next location.
>
> Thanks for the suggestion. It’s not 100% clear to us what an oracle that “pattern-matched” on position given edit symbol would look like, but we agree it’s interesting to think about. To answer the larger question of what the hardest part of the prediction is, we include some additional plots in Appendix B.3 of an updated version. The plots show that the most difficult part is predicting the jumps when one pattern is completed and then the model must decide where to edit next.
>
> > The explicit baseline model performs very well on all the edit types in Table 1. Are there cases where even this explicit case works poorly? Is the improved implicit model *always* upper bounded by the explicit model (to me it seems like the answer should always be yes, but it would be interesting to check it out for cases where explicit is not very high accuracy).
>
> The implicit model gets slightly better results on the “MultiTask” problem in Table 1, so the explicit model is not *always* more accurate. However, we did find the explicit model to produce good accuracy across the board. The main issue with it is the memory and computational cost.

---

### Official Review · AnonReviewer2 · 2018-11-03
**Good baselines for a new task**

**Rating:** 6
**Confidence:** 4

**Review:**

The paper provides good baselines for predicting edits in a text (evaluated on source code) learned from a history of changes. This is an interesting problem that has not beed systematically studied in literature, with the exception of several sporadic works from the software engineering community. Fully predicting the code text writing process as opposed to the code itself is an interesting task with possible big impact, IF the accuracy of this edit model manages to significantly outperform simple left-to-right code text prediction techniques.

One of closest related works to this A somewhat similar system with language models for predicting APIs based on sequences is [1], it would help to compare to it at least on a conceptual level. is [2] that predicts if a "change" is complete i.e if it misses to complete a change. However it does not predict the entire edit process, but only the last missing piece of a change (usually a bug if it is missed).

Pro:
 - First fine grained text (code) evolution evaluation and formulation of the challenge to provide labels for predicting the process of code writing.
 - Discussion about effectiveness of the introduced explicit vs implicit tasks and good trade-offs discussion.
 - Techniques are applicable and interesting beyond code.
 - Well-written and easy to follow text

Against:
 - The introduced models are based on simple sequence representations of code tokens. Once more semantic representation (let's say ASTs) are taken, the implicit attention techniques may need to also be updated.
 - It would help to elaborate more on the process defined in 5 for the real dataset that given a pair of code snapshots, assumes that one is obtained by simply inserting/removing the tokens in sequence. In particular, it could be that a candidate model predicts a different sequence with the same results in the snapshots. Then the training loss function should not penalize such solutions, but it will if the sequence is not strictly left-to-right. Did you need to tweak something here especially for larger changes?

[1] Anh Nguyen, Michael Hilton, Mihai Codoban, Hoan Nguyen, Lily Mast, Eli Rademacher, Tien Nguyen, Danny Dig. API Code Recommendation using Statistical Learning from Fine-Grained Changes
[2] Thomas Zimmermann, Andreas Zeller, Peter Weissgerber, Stephan Diehl. Mining version
histories to guide software changes.

---

> ### Author Response · Authors · 2018-11-26
> **Response to "Good baselines for a new task"**
>
> Thank you for your review and the additional references.
>
> > Once more semantic representation (let's say ASTs) are taken, the implicit attention techniques may need to also be updated.
>
> This was a common question amongst reviewers, so we responded to it in our general response above. While we agree the encoder would need to be updated to use more structured representations of the code, the rest of the formulation would be applicable with minimal changes. This is a good direction for future work.
>
> > It could be that a candidate model predicts a different sequence with the same results in the snapshots. Then the training loss function should not penalize such solutions, but it will if the sequence is not strictly left-to-right. Did you need to tweak something here especially for larger changes?
>
> There are of course situations where the imposed left-to-right order is not the true sequence of edits that was used to transform between snapshots by the developer. However, the snapshots are collected often during development (far more frequently than typical git commits), and so the number of edits between snapshots is usually small. Additionally we filter out edit sequences when the number of edits between two snapshots is large. As such, while the edit sequence we impose between pairs of snapshots may not be fully correct, we believe it is a good approximation and rarely too far from the real edit sequence.
>
> We readily acknowledge that this strategy of assuming left-to-right order for edits between snapshots would likely not work as well for data with larger changes between snapshots like in git commit histories. In this case, one option would be to explore an EM-like training procedure, where the ordering is a latent variable that is inferred during training. However, in this work we focus on developing approaches for the fully supervised task and leave such extensions for future work.
>
> > A somewhat similar system with language models for predicting APIs based on sequences is [1], it would help to compare to it at least on a conceptual level. is [2] that predicts if a "change" is complete i.e if it misses to complete a change. However it does not predict the entire edit process, but only the last missing piece of a change (usually a bug if it is missed).
>
> Thank you for pointing us to these references, they are certainly very relevant! There are a number of similarities, as well as important differences, between our problem formulation and approach compared to these. We actually do cite [2] in the paper, but we will elaborate on the differences with our work. With respect to [1], the problem is to predict a method call given a set of preceding changes and a location. That is, their approach assumes that the location is given, and are restricted to method calls from a given API. In [2], they are instead predicting the location of the next change, given the location of a previous change. Both of these utilize co-occurrence frequencies between changes in order to provide a ranking of suggested changes/locations respectively. In our approach, we try to predict both location and change at a finer level of granularity (subword tokens), using the entire sequence of changes up until the current timestep. This allows the model to potentially generalize previously seen edit patterns, to new locations in the code (as in the synthetic tasks), and helps to resolve ambiguities based on user intent (as in Figure 1). We will happily include these references in our discussion on related work.

---

### Official Review · AnonReviewer3 · 2018-11-05
**Promising work for a new task**

**Rating:** 6
**Confidence:** 2

**Review:**

Although the subject of the task is not quite close to my area and the topic of programming language edit is relatively new to me, it is a comfortable reviewing for me thank to the well-written paper. This paper formalize a new but very interesting problem that how to learn from and predict edit sequence on programming language.

Pros:
+ This paper proposed two different data representation for the tokens in text, implicit and explicit. The paper starts with a very interesting problem of programming language editing, in that the intend of source code developer's intent is predicted.
+ The two different representations are well described, and dis/advantages are well elaborated.
+ The writing is very technical and looks solid.
+ Both synthetic dataset and real source code dataset are exploit to evaluate the performance of all models proposed.

Questions:
1.	The content of text supposed to be programming language, but neither the model design nor synthetic dataset generation specify the type of text.
2.	Further, if the text type is specified to be source code, I suppose each token will has its own logical meaning, and a line of source code will have complex logic structures that is not necessarily a flat sequence, such an “if…else…”, “try…catch…”, “switch…case…” etc. How do you address this issue with sequence models such as LSTM?
3.	In generating synthetic dataset, where is the vocabulary from?

Minor issues:
1.	The first sentence in Abstract doesn’t look right: “Programming languages are emerging as a challenging and interesting domain for machine learning”. I suppose you meant: “Programming language generation/edit….”

---

> ### Author Response · Authors · 2018-11-26
> **Response to “Promising work for a new task”**
>
> Thank you for your thoughtful review. Responses to your questions are below.
>
> > The content of text supposed to be programming language...
>
> This was a common question amongst reviewers, so we responded to it in our general response above.
>
> > source code will have complex logic structures that is not necessarily a flat sequence, such an “if…else…”, “try…catch…”, “switch…case…” etc. How do you address this issue with sequence models such as LSTM?
>
> There is some evidence that LSTM models can reasonably learn more complex structures (ftp://ftp.idsia.ch/pub/juergen/L-IEEE.pdf, https://arxiv.org/abs/1412.7449, https://arxiv.org/pdf/1805.04908.pdf), but ultimately we do think it will be worthwhile to move away from flat models of code within the edits formulation, analogously to how recent works are moving towards graph-structured representations of static snapshots of code (e.g., https://arxiv.org/abs/1711.00740). See the discussion about encoder structure in the general response above. We believe it’s relatively straightforward to combine the edits formulation we present here with alternative, more structured models of source code.
>
> > In generating synthetic dataset, where is the vocabulary from?
>
> As described in 6.1, we use a vocabulary with a fixed size. Internally, the vocab elements are just represented by their integer indices. The strings associated with the vocab elements are arbitrary, but for compactly displaying them, we just use the sequence of capital letters (A gets index 0, B gets index 1, …).

---

### Official Review · AnonReviewer4 · 2018-12-07
**Great start for the novel task of modeling source code edits**

**Rating:** 5
**Confidence:** 4

**Review:**

The paper presents a very interesting new task of modeling edits to a piece of source code. The task also has immediate practical applications in the field of Software Engineering to help with code reviews, code refactoring etc.

Pros:
1. The paper is well-written and easy to follow.
2. The task is novel (to my knowledge) and has various practical applications.
3. Many obvious questions that would arise have been answered in the paper for eg., the contrast between explicit and implicit data representations.
4. The use of synthetic data to learn about the types of edits that the models can learn is a very good idea and its inclusion is much appreciated.
5. Evaluation on a very large real dataset demonstrates the usefulness of the model for real world tasks.

Cons:

In general, I really like the task and a lot of the models and experiments, but the description of the real world experiments is severely lacking in information and results. Also, there are many unanswered questions about the synthetic experiments.

1. Firstly, where is the data obtained from? Who are the programmers? What was the setting under which the data was collected in ?
2. The paper doesn't provide examples from this real world dataset nor are there examples of model predictions on this dataset.
3. What are the kinds of edits on the real world dataset? What happens if someone adds a 100 lines to a file and saves it? How is this edit added to the dataset?
4. Some Error analysis on the real world data? It's hard to understand how the model is doing by just reporting the accuracy as 61.1%. Lots of the accuracy points maybe obtained for obvious commonplace edits like keywords, punctuation etc.. ?
5. Some more dataset stats related to the real world data. For eg., how many tokens are in each snapshot?
6. "We evaluate models on their ability to predict a sequence of subtokens up to the next token boundary" <-- Require more details about this. This section needs more clarity, its hard to interpret the results here.
7. Are you going to release this real world dataset?
8. If the real world dataset is large, why don't you report running time stats on that? Why use the synthetic dataset for testing scalability? If you have 8 million edits, that seems like a big enough dataset. How long did your models take on these?

Re: the synthetic dataset

1. It's nice that you've tested out different single synthetic edit patterns, but what happens in a dataset with multiple edit patterns? Because this will be the scenario in a real world dataset.
2. What happens when the order of edits is different in the prediction but the outcome is the same? Suppose we have edit A followed by B and the model predicts B followed by A, but the end result is the same? The accuracy metric will fail here? How can this be addressed?

---

> ### Author Response · Authors · 2018-12-10
> **Response to "Great start for the novel task of modeling source code edits"**
>
> Thank you for your review and for your questions.
>
> > 1. It's nice that you've tested out different single synthetic edit patterns, but what happens in a dataset with multiple edit patterns? Because this will be the scenario in a real world dataset.
>
> This is already in the paper. See the “MultiTask” dataset in Section 5 and results in Table 1.
>
> > 2. What happens when the order of edits is different in the prediction but the outcome is the same? Suppose we have edit A followed by B and the model predicts B followed by A, but the end result is the same? The accuracy metric will fail here? How can this be addressed?
>
> The task as defined in the paper is to predict the order of edits, so A-then-B is considered different from B-then-A, which we think is sensible if the target task is suggesting next edits to a developer. We don't think a tool would be very useful if it was allowed to suggest any edit that a developer would make in the future. It could force the developer into a lot of unwanted context switching. However, please see the response to Reviewer 2 for an answer to a related question about alternative loss functions related to ordering issues.
>
> > 1. Firstly, where is the data obtained from? Who are the programmers? What was the setting under which the data was collected in ?
>
> This is left vague for anonymity. The data set is from professional developers doing their day-to-day work. We will include de-anonymized details in the final version.
>
> > 3. What are the kinds of edits on the real world dataset? What happens if someone adds a 100 lines to a file and saves it? How is this edit added to the dataset?
>
> Thanks for asking this. We should have noted that we require edit sequences to have at least three snapshots and no more than 256 subtoken changes. So the edit you describe will be pruned from the dataset.
>
> > 5. Some more dataset stats related to the real world data. For eg., how many tokens are in each snapshot?
>
> We say in 6.2 that the initial states have up to 1000 subtokens in the snapshots.
>
> > 6. "We evaluate models on their ability to predict a sequence of subtokens up to the next token boundary" <-- Require more details about this. This section needs more clarity, its hard to interpret the results here.
>
> Could you clarify what additional details you'd like to see?
>
> > 7. Are you going to release this real world dataset?
>
> Unfortunately this isn't possible due to the sensitive nature of the data. However, we have been discussing possibilities for releasing a small sample or more creative solutions (e.g., predicting imputations of finer-grained changes from coarser-grained GitHub commits). If there is interest from the community and the paper is accepted, this is a discussion that we’d be interested in having at the conference.
>
> > 8. If the real world dataset is large, why don't you report running time stats on that? Why use the synthetic dataset for testing scalability? If you have 8 million edits, that seems like a big enough dataset. How long did your models take on these?
>
> The synthetic data allowed us to vary the relevant dimensions and evaluate running times in a more controlled way. There is nothing significantly different about the real data. We’ll add precise numbers to an updated draft, but on the real data, the implicit attention model took roughly 6 hours to reach maximum validation accuracy, the implicit RNN model took roughly 10 hours, and the explicit model took roughly a week.
>
> > 4. Some Error analysis on the real world data? It's hard to understand how the model is doing by just reporting the accuracy as 61.1%. Lots of the accuracy points maybe obtained for obvious commonplace edits like keywords, punctuation etc.. ?
> > 2. The paper doesn't provide examples from this real world dataset nor are there examples of model predictions on this dataset.
>
> This will take some work and isn’t something that we can turn around immediately, but if the paper is accepted, we will commit to adding example model predictions and further error analysis to the final version.

---

### Author Response · Authors · 2018-11-26
**General response to all reviewers**

Thank you to all of the reviewers for your thoughtful reviews. All reviewers were generally positive about the paper, especially with regard to it presenting an interesting and novel problem formulation with significant potential impact.

The main common line of questioning among the reviewers was whether it would be fruitful and straightforward to consider extensions that incorporate code structure, such as an abstract syntax tree (AST) representation.

The extension is straightforward for the implicit attention model, although there are some design questions that would need to be studied. For data preparation, it is possible to compute diffs between tree-structured representations of snapshots using a tool like Gumtree (https://github.com/GumTreeDiff/gumtree), in which case the edits to trees could be broken down into insertions and deletions with locations represented as pointers to nodes in the tree. If you look at Figure 3, the decoder could work unmodified assuming that “implicit positions” are replaced with something like “node id” in a tree or graph. Thus, the main question would be how to design the encoder to jointly represent tree- or graph- structured initial state & previous edits. Natural starting points would be tree- or graph- neural networks, or if one wanted to stay within the Transformer framing of the problem, something like relative self-attention could be promising (https://arxiv.org/abs/1803.02155). Another structural idea that might be nice to explore in future work could be attention over subtrees rather than just nodes, as in https://arxiv.org/abs/1802.03691.

So while we see more structured encoders as a natural extension and ultimately where we want to go with this line of work, our intention in this paper is to stay focused on formulating the problem of learning from and generating edits, and answering some of the more basic cross-cutting questions (e.g., should we use explicit vs implicit models?), leaving further architectural exploration for future work. This view also potentially allows the approach to be applied more generally to other domains involving edits to language, though fleshing out other real applications in other domains could be a large undertaking and is left for future work.

In summary, this paper motivates and defines a new way of thinking about models for generating source code. Reviewers agree with the novelty and importance of the problem (R1: “The application is obvious … would be very valuable.”, R2: “interesting task with possible big impact”, R3: “very interesting problem”). The paper lays out several modeling approaches, compares their speed and accuracy on synthetic data and a real application, and discusses tradeoffs. Reviewers appreciate these components (R1: “the idea being to match the performance of the explicit (heavy) model with the implicit model. This is shown to be challenging, but a clever model is introduced that achieves this, and is thus the best of both worlds.”; R2: “good trade-offs discussion”’  R3: “dis/advantages are well elaborated”). The other major questions are mostly about modeling extensions (more structured encoders; adding latent variables to represent ordering) or in other applications (text editing), which we take to be a sign that there are many interesting ways to build on this work.

We will provide answers to detailed reviewer-specific questions as direct replies to the individual reviews.

---

### Meta-Review · Area_Chair1 · 2018-12-14
**the overall substance and novelty are marginal.**

**Confidence:** 2
**Recommendation:** Reject

**Metareview:**

This paper focuses on neural network models for source code edits. Compared to prior literature that focused on generative models of source codes, this paper focuses on the generative models of edit sequences of the source code. The paper explores both explicit and implicit representations of source code edits with experiments on synthetic and real code data.

Pros:
The task studied has a potential real world impact. The reviewers found the paper is generally clear to read.

Cons:
While the paper doesn't have a major flaw, the overall impact and novelty of the paper are considered to be relatively marginal. Even after the rebuttal, none of the reviewers felt compelled to increase their score. One point that came up multiple times is that the paper treats the source code as flat text and does not model the semantic and syntactic structure of the source code (via e.g., abstract syntax tree). While this alone would have not been a deal-breaker, the overall substance presented in the paper does not seem strong. Also, the empirical results are reasonable but not impressive given that the experiments are focused more on the synthetic data, and the experiments on the real source code are weaker and less clear as has been also noted by the fourth reviewer.

Verdict:
Possible weak reject. No significant deal breaker per say but the overall substance and novelty are marginal.